

# Close range photogrammetric methods applied to the study of the fronts of Johnsons and Hurd Glaciers (Livingston Island, Antarctica) from 1957 to 2013.

Ricardo Rodríguez[1], Julián Aguirre[2], Andrés Díez[2], Marina Álvarez[3], Pedro Rodríguez[4].

(1) Departamento de Señales, Sistemas y Radiocomunicaciones. ETSI de Telecomunicación. Universidad Politécnica de Madrid.
(2) Departamento de Ingeniería Topográfica y Cartografía. ETSI en Topografía, Geodesia y Cartografía. Universidad Politécnica de Madrid.
(3) Departamento de Lenguajes y Sistemas Informáticos e Ingeniería de Software. ETS de Ingenieros Informáticos. Universidad Politécnica de Madrid.
(4) Departamento de Matemática Aplicada. ETSI de Telecomunicación. Universidad de Málaga.

## Abstract

The study of glacier fronts combines different geomatics measurement techniques as the classic survey using total station or theodolite, technical GNSS (Global Navigation Satellite System), using laser-scanner or using photogrammetry (air or ground). The measure by direct methods (classical surveying and GNSS) is useful and fast when accessibility to the glaciers fronts is easy, while it is practically impossible to realize, in the case of glacier fronts that end up in the sea (tide water glaciers). In this paper, a methodology that combines photogrammetric methods and other techniques for lifting the
front of the glacier Johnsons, inaccessible is studied. The images obtained from the front, come from a non-metric digital camera; its georeferencing to a global coordinate system is performed by measuring points GNSS support in accessible areas of the glacier front side and applying methods of direct intersection in inaccessible points of the front, taking measurements with theodolite. The result of observations obtained were applied to study the temporal evolution (1957-2014) of the position of the Johnsons glacier front and the position of the Argentina, Las Palmas and Sally Rocks lobes
front (Hurd glacier).

**Link to the data repository:** http://doi.pangaea.de/10.1594/PANGAEA.845379

## Study area and previous works

The Johnsons and Hurd glaciers are situated in Hurd Peninsula (Navarro et al., 2011), southwest of Livingston Island, located in turn in the South Shetland Islands of Antarctica. The Johnsons glacier ends at sea and produce icebergs (tide
water glacier), changing its shape continuously; in the central area of the forehead, above 50 meters per year speeds, with a height of over 50 meters (Rodriguez, 2014) are reached. Not so with Hurd glacier fronts (Argentina, Las Palmas and Sally Rocks) that end up in land and therefore do not produce icebergs landslides (see Figure 1). This makes its way not change so significantly in a short space of time, with speeds below 5 meters per year in the central part of the fronts (Rodriguez, 2014).

The previous work carried with data collection fronts under study are summarized as follows:

- DOCU 1: Flight made by the British Antarctic Survey (BAS) in December 1957. A total of 5 frames (X26FID0052130, X26FID0052131, X26FID0052132, X26FID0052160 and X26FID0052161) are selected to study all of the glacier fronts.
- DOCU 2: Flight made by the United Kindom Hydrographic Office (UKHO) in January 1990. A total of 3 frames
(0097, 0098 and 0099) are selected for the study of the entire glacier fronts.
- DOCU 3: Photogrammetric survey (metric camera) from the top of the glacier front Johnsons in 1999 by the University of Barcelona (Palà et al., 1999).



- DOCU 4: Aerial photograph obtained by the Quickbird system in January 2010 for the Hurd Peninsula.
- DOCU 5: Aerial photograph obtained by the Quickbird system in February 2007 for the Hurd Peninsula.
- DOCU 6: Inventory of data (2000-2012) by the Group of Numerical Simulation in Science and Engineering of the Polytechnic University of Madrid (GSNCI). These observations are made with GNSS techniques and theodolite and exclude the position of the glacier front Johnsons.
- DOCU 7: Photogrammetric survey (non-metric camera) of the front wall of Johnsons Glacier conducted in February 2013.

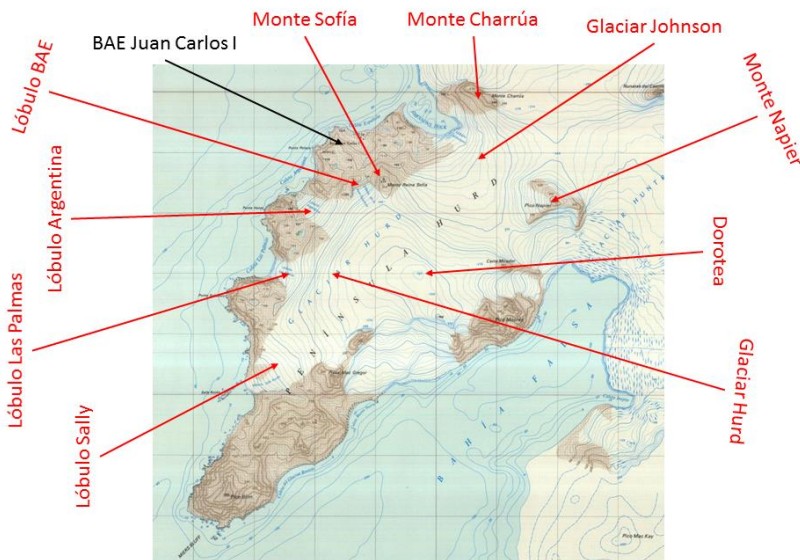

*Figure 1. Situation of the Johnsons and Hurd glaciers, location of the major landforms and situation of the Spanish Antarctic Base Juan Carlos I. Base map scale 1: 25000 Geographic Service of the Army in 1991.*

## The Photogrammetry

Photogrammetry is the science or art including those methods used to determine the geometrical properties and the spatial location of objects, from the interpretation and measurements made in two or more of these photographs, taken from different viewpoints and with some common parts. These methods all objects that could be photographed, as in the present case, apply a glacier front. The image registration involves the extraction of reality, without the need to physically cross it for data collection and the possibility of subsequent study in office, without relying on bad weather, which is quite important in these settings. That further work will be to obtain three-dimensional information from two-dimensional featuring photography, thanks to the stereoscopic vision provided by two different points of view (Wolf, 1983).

Photographic images used in photogrammetry are affected by a number of systematic errors due to instrument used and the environment in which it has made the photo shoot. In aerial photogrammetry latter are important because the distance between the ground and the photo sensor is large. However in terrestrial photogrammetry, since the much closer to the position of the object takes, these errors decrease and become more important due to the instruments used. The calibration process is used to calculate a number of parameters inherent in the camera or internal parameters (focal the calibrated, the position of the principal point of symmetry or PPS function and distortion) that allow knowing the geometry of the beam and thus perspective, to correct these systematic errors of the images (Scherz, 1974).

The camera model "pin-hole" is generally adopted as the basis for geometric modeling cameras (Young, 1989). Explain how a 3D point is projected onto the 2D image plane of a camera. It is a simplification of reality, since calibrate a real



model would be a great deal of time and cost and the result is quite acceptable, though not exact. This transformation of coordinates 3D terrain 2D image coordinates is determined by solving the internal parameters of the camera.

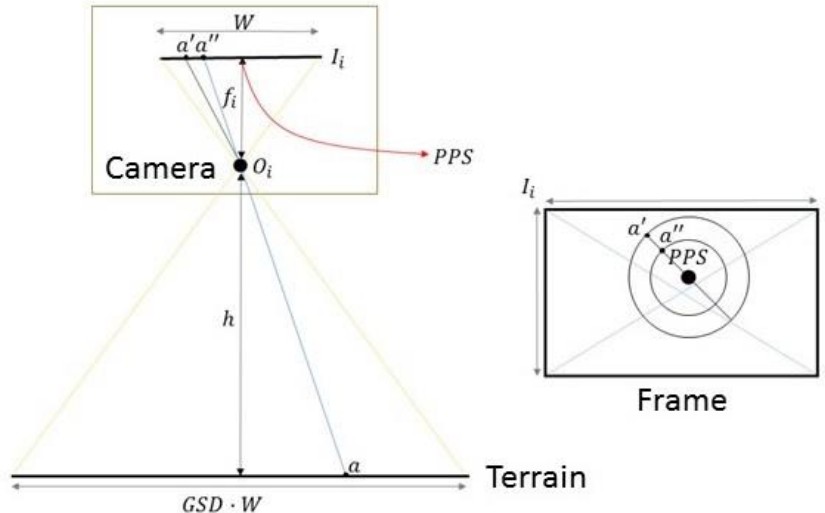

*Figure 2. The point field $a$ projects ideally in the camera sensor according to $a''$ although due to the radial distortion that occurs projection is $a'$ ($\Delta r = a' - a''$). GSD (Ground Sample Distance) is defined as the pixel resolution on the ground, being $f_i$ the camera focal, $h$ the distance that the object to be photographed is.*

To carry out the process has to start from a geometric camera model and a functional or mathematical model that explains the formation of the image. Derived from the functional model of collinearity condition (shown later) and applying the projective transformation (Shan, 1999). This transformation is usually not sufficient because ideally, the rays of light from the scene should go through the optical center linearly, but in practice, the lens systems are composed of several elements,

introducing a nonlinear distortion the optical paths and the resulting images. The distortion varies with the distance of each point to the center of the optical axis. The most commonly used approach is to decompose to correct distortion in their radial components (Figure 2) and tangential (Brown, 1971). In practice, the radial distortion $\Delta r$ is much larger than the tangential distortion so that only the first, which is expressed by the equation [1] for a point coordinate image (x, y) is ignored.

$$\Delta r = k_1 r^3 + k_2 r^5 + k_3 r^7$$
$$r = \sqrt{(x - x_0)^2 + (y - y_0)^2},$$

[1]

where $k_i$ are the coefficients of radial distortion $(x_0, y_0)$ are the coordinates of PPS in the image plane and $\Delta r$ is the radial distortion to the point considered.

Once calculated systematic errors and how to correct them and to obtain three-dimensional information from two-dimensional featuring photography, photogrammetry part of the solution that provides stereoscopic vision, in which one stage photographically recorded from two different view with a common coating, three-dimensionally can be played

directly through spatial intersection producing each pairing of homologous rays appearing in both images. The procedure would be the following (Figure 3):



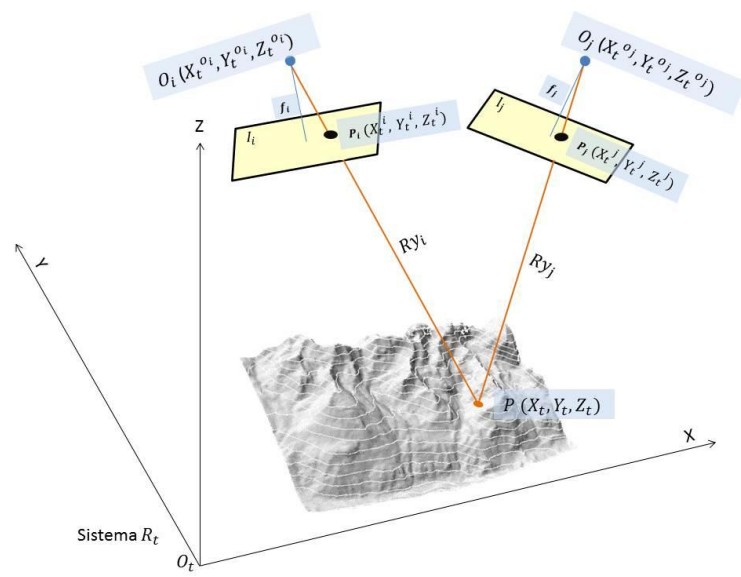

*Figure 3. Representation of point P in each of the photographs. Thus, one can calculate the unknown point coordinates in a reference system from flat photographs taken from two different points of view.*

- A terrestrial reference system $R_t$ with $O_t$ center, which is the surface you want to measure and containing the point $P$ with coordinates $(X_t, Y_t, Z_t)$ in this reference system is fixed.
- Two photographs are made from different viewpoints in this surface from $O_i\left(X_t^{O_i}, Y_t^{O_i}, Z_t^{O_i}\right)$ and from $O_j\left(X_t^{O_j}, Y_t^{O_j}, Z_t^{O_j}\right)$ (referred to $R_t$ coordinate reference system).
- Point P is represented in each of the photographs for their projections $P_i\ (X_t^i, Y_t^i, Z_t^i)$ and $P_j\ (X_t^j, Y_t^j, Z_t^j)$ (referred to $R_t$ coordinate reference system).

All this can be expressed by the so-called collinearity conditions (Kraus, 2000) which states that the center of projection, an image point and corresponding on the ground, are in the same line. The equations determining the point $P$ (whose coordinates are intended to determine), the point $P_i$ and the center of projection $O_i$ are in the same line. In like manner, the point $P$, the point $P_j$ and the center of projection $O_j$ are collinear. From the above it follows that the points $P$, $P_i$, $P_j$, $O_i$ y $O_j$ are all in the same plane, allowing us to calculate the coordinates of any point $P$ in the reference system $O_t XYZ$, by applying transformations spatial similarity (Kraus, 2000).

**Photogrammetry with non-metric camera**

To take measurements Johnsons Glacier (February 2013), a non-metric DSLR (Digital Single-Lens Reflex) camera used. The choice of this type of camera can cut costs without excessive loss of accuracy. Obviously, its implementation requires a photogrammetric recalibration, which allow to know with sufficient accuracy the internal geometry thereof. One of the parameters to be determined in the calibration process will be the role of lens distortion [1], which systematic errors be corrected captured images. The next step will be the photogrammetric orientation, giving a stereoscopic model of the area covered by both images, for later reference it to the ground coordinate system chosen. For this purpose a series of control points of measured field is used by GNSS techniques and classical topography. After this phase, it is able to extract the graphic elements of interest from the glacier front by stereoscopic restitution.

Field work in front of the glacier, consisting of measuring checkpoints and photo shoot. Due to the dynamic nature of the scenario we face, should be nearly simultaneous tasks, as if among the topographic measurement and photographic takes a large time period and therefore movement of the front, errors will occur in the spatial referencing of pictures taken.



a) Photogrammetric support

We call the set of photogrammetric control points. Establishing a network realizing permanent bases enabling the system in the field, in this case, the projection 20S UTM on the WGS84 ellipsoid to the coordinates of these points are adopted in the reference system is necessary.

The work was conducted in February 2013, with fewer clouds, high visibility in the area and windless conditions had to be met to ensure the goodness of the observations. The measurement of the control points was performed using GNSS techniques, using the Trimble 5700 GPS from that provided in the Spanish Antarctic Base Juan Carlos I. Three bases (B1000, B2000 and B3000) were established, as can be seen located in Figure 4. In addition, six control points were measured by the same technique at the ends of the front of the glacier, being accessible on foot (points P100, P200, P300, P400, P500 and P600). These points are materialized on the ground with red flags to be reachable in photographs. GNSS mode was static work with a minimum relative parking time fifteen minutes later verifying with direct visual including theodolite, using the coordinates of the reference base as B1000.

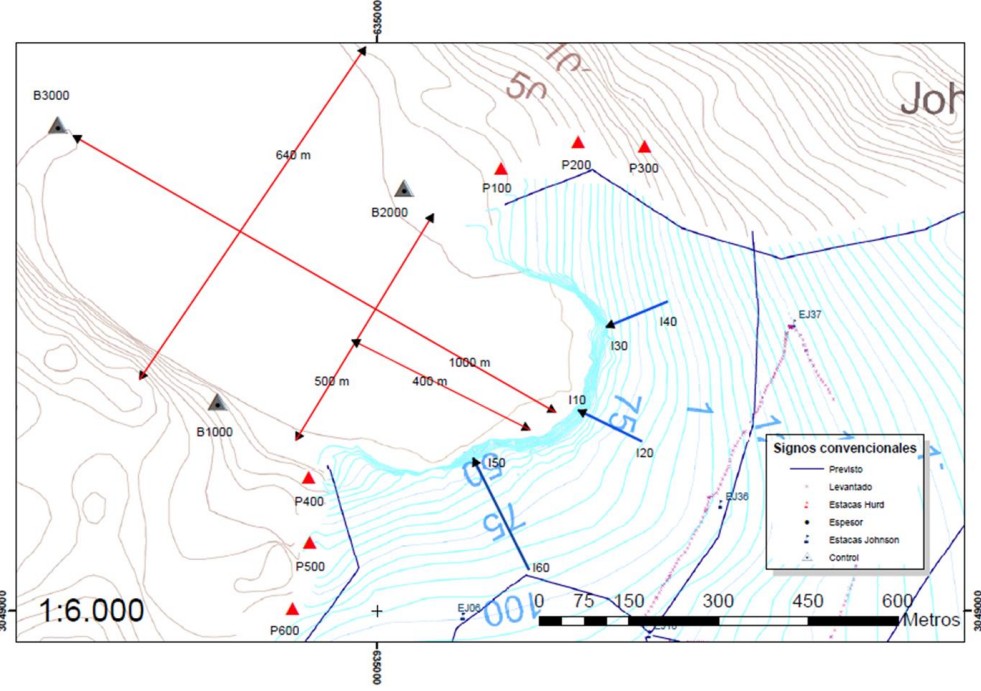

*Figure 4. Mapping of the B1000, B2000 and B3000 bases from which theodolite measurements were performed. In red triangles, control points measured using GNSS techniques and black triangles control points on the glacier front Johnsnons obtained by direct intersection.*

That same day a number of points were measured on the glacier front in order to obtain the same field coordinates and place as control points to add to those already obtained by GNSS. In this case were made visually each from three bases (B1000, B2000 and B3000) using an optical theodolite Wild Heerbrugg T1 available in base Juan Carlos I, in order to avoid problems with the electronics due to moisture and temperature. Previous picture of the front of the glacier, where the points were to observe and to identify them from all databases seamlessly labeled was used for this work. Theodolite such does not allow the measurement of distances, thereby direct intersection methodology used for angle measurements,





more specifically, by resection from the three bases, allowing to have some redundancy in the observations (Domínguez, 1993).

When measuring points from different bases appeared a complication that was not initially planned. Due to the change in perspective of the support points taken from the first season, they were not easily visible from the rest of the bases with many of the points provided could not be measured by resection and therefore were discarded. However, thanks to the spots that produced the different volcanic deposits, was identified and measured five points on the glacier front (i20, i30, i40, i50 and i60).

b) Shooting

As already mentioned, the camera used is a DSLR camera (Nikon D60 with lens 55-200mm AF-S DX 10 megapixel). Sockets with parallel and perpendicular to the front of the glacier and appeared in all the glacier front Johnsons, from the top line to the bottom of contact with water intakes optical axes in the normal case, ie projected. To get only they could make the photos from the water, so he decided to do them from a zodiac. Two parallel glacier front passes are performed; the first at a distance of approximately 400 m, with a focal length of 95 mm and focus to infinity. The second was conducted at a distance of 700 m from the glacier front with the same focal length and the same approach. The coating was superior to 95%, since an excess of photographs to filter them later in cabinet and keep the shots better ensured a coating of 80%, in addition to having good quality is preferred.

To solve the problem of reducing the number of control points for not having complied with the resection and to get footholds in the front, it was decided to conduct two series of converging photos from a high area near the glacier and to provide and coordinate a number of points to complete the supports. These photographs were taken with a focal length of 130 mm and focus to infinity. It was also observed that in these photographs appeared stakes identified points on the glacier (measured with GNSS techniques) that had been previously fitted with ground coordinates (Navarro *et al.* 2013), the EJ06, EJ36 and EJ37 stakes (Figure 4).

c) Calibration

Calibration is the first phase taking place in the office, before or after making the photographic field, but respecting the same parameters as was done that. If a previous calibration is chosen, it is forcing decision-making photographic field with the already calibrated parameters and not always for the environment and lighting conditions we found there is achieved. However, performing then there are no major problems playing the same exposure, the same parameters used in the decision.

To carry it out, as a pattern to photograph the building facade "Mirador", located on the street Princesa de Eboli in Madrid, which was ideal configuration for the project was chosen. This building also has a large open space at the front, allowing a stand similar to that used in photographic shots from the glacier front Johnsons (400 meters) away; this distance also makes it easy to measure total station the corners of the windows that will be needed in the calibration grid. For forming the corners of the reticle windows situated in a lower horizontal line, an upper horizontal line, two central horizontal lines and three vertical lines leaving a dot pattern as can be observed in Figure 5 were chosen.

First survey data all chosen points was performed using the total station. Because the instruments used did not allow reflectorless measuring distances over 100 m, we chose to determine the coordinates of points using only angular measurements, applying the methodology of direct intersection (Dominguez, 1993). Observations were made from two stations simultaneously, providing one of arbitrary local coordinates for the other station coordinates and all grid points. The coordinates of the second station were obtained with an error of 12 mm. In this process the calculator "Leica Geoffice" that yielded the local coordinates of all grid points with a mean square error of 53 mm, eliminating the points whose residues exceeded this magnitude is used.





Then taking photographs was performed with focal of 85, 95 and 130 mm (those used in decision glacier front). With the images and the actual coordinates of the building points obtained with total station is able to estimate the internal parameters of non-metric camera used.

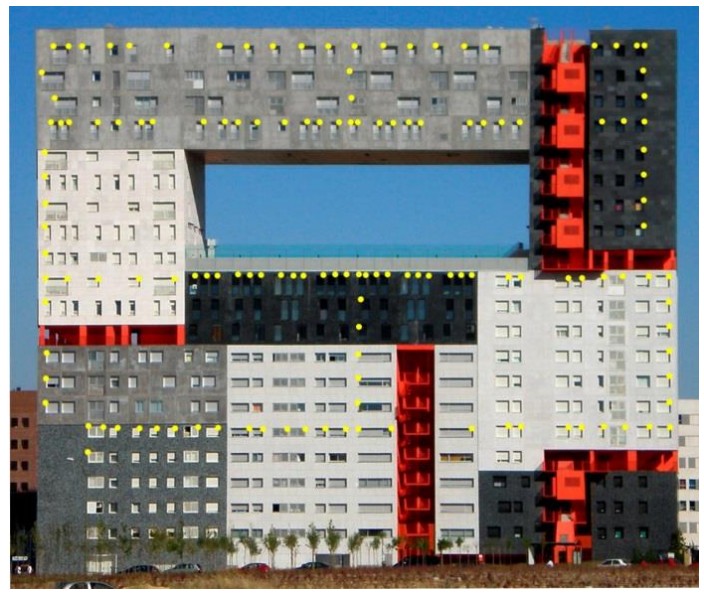

*Figure 5. View of the facade of the building Mirador used for the calibration of non-metric camera used in photographic shots of Johnsons glacier front. Yellow, the measured points by using a total station.*

In this case, the results for the calibration function were as follows (see equation [1]):

- For a focal of 85 mm: $x_0 = 1889\ px$; $y_0 = 753\ px$; $k_1 = 1.16682388530791E - 08$; $k_2 = -4.82380671349918E - 15$; $k_3 = 5.49321773632122E - 22$.
- For a focal of 95 mm: $x_0 = 2043\ px$; $y_0 = 602\ px$; $k_1 = 1.18819659796645E - 08$; $k_2 = -4.07129365804265E - 15$; $k_3 = 3.78566248003642E - 22$.
- For a focal of 135 mm: $x_0 = 2019\ px$; $y_0 = 629\ px$; $k_1 = 1.46381151286755E - 08$; $k_2 - 6.20810960238916E - 15$; $k_3 = 7.13856817510912E - 22$.

c) Preparing images

One of the drawbacks of work in extreme environments is that they do not allow the repetition of field observations if subsequent errors are detected in office. This makes copper the very relevant fact take as much information as possible during observations, so in each of the past photo shoots excess were performed to ensure quality and continuity of data. That is why the first thing to realize is a selection of the necessary photographic peer models covering approximately 80%.

A particular operation in the process of this project has been to correct distortion all photographs using internal orientation parameters and generating a new set of corrected images of distortion (see Figure 6), so that may be brought into any photogrammetric program without matter what type of distortion model used.



d) Calculation of ground coordinates

With these photographs as a starting point (free photographic shots of distortion), collinearity conditions apply, using the known support points obtained in the process of photogrammetric ground coordinates to obtain the parameters of different similarity transformations which can obtain the ground coordinates of any point of the photographs.

5    The results to be correct, with absolute waste 0.7 m below ground coordinates, taking into account the accuracy of topographical measurements of the bearings, each about 0.5 m.

Once all parameters calculated from different similarity transformations, photographs are introduced together with these parameters in a software developed for this purpose that allows the photogrammetric restitution from the glacier front Johnsons, without artificial stereoscopic vision (see Figure 7).

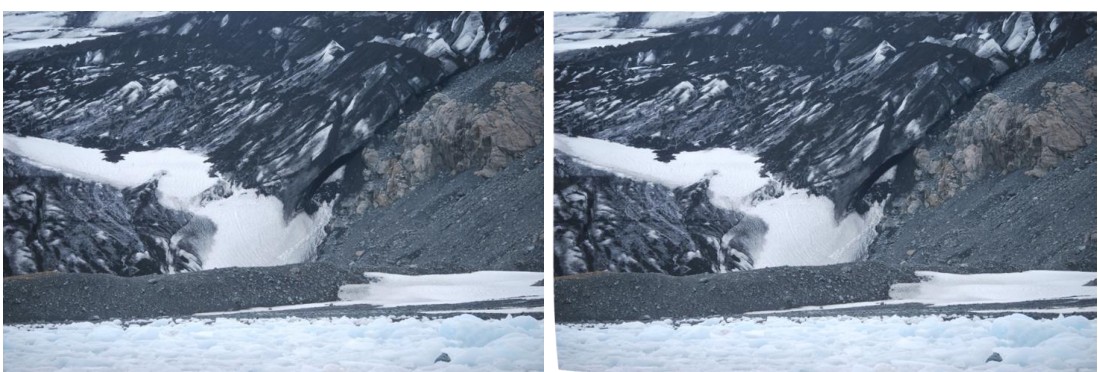

10    *Figure 6. On the left, the original image obtained with no metric camera. On the right, the rectified image after applying the distortion function. It can be seen in the lower left corner of the two images, an area affected by the radial distortion.*

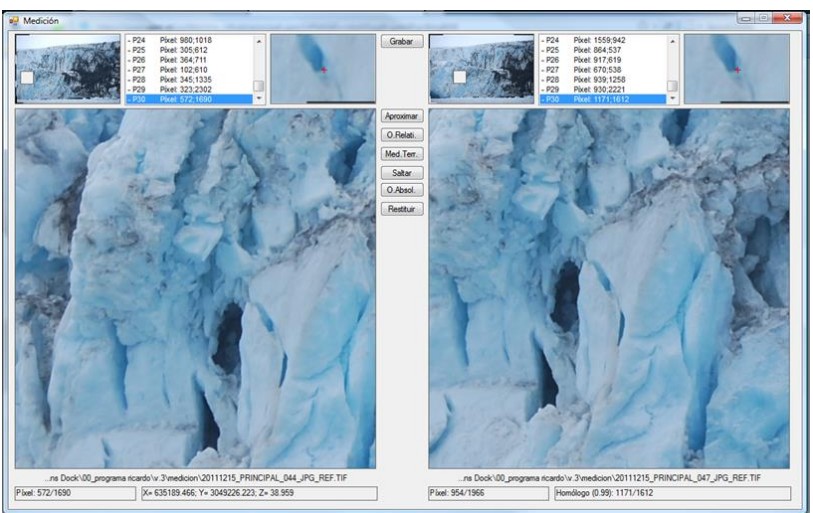

*Figure 7. Main screen of the software developed for photogrammetric restitution. Such software does not require artificial stereoscopic vision and for the restitution by locating points in both images. Clicking on an item in the frame on the left, locate the point right frame of automatic correlation.*



This glacier front mainly concerned the top line and the bottom line (intersection with the sea) thereof. And also since this presents different strata glacial ash, the main lines defining each of these strata. This process is done using the software developed. The results of this refund, you can see in Figure 8 (top), with a maximum error of 0.7 m as mentioned above. With a total of 180000 points returned by automatic correlation, orthophoto shown in Figure 8 (below) was also obtained.

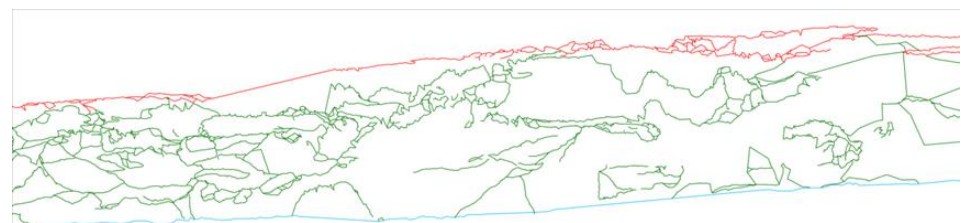

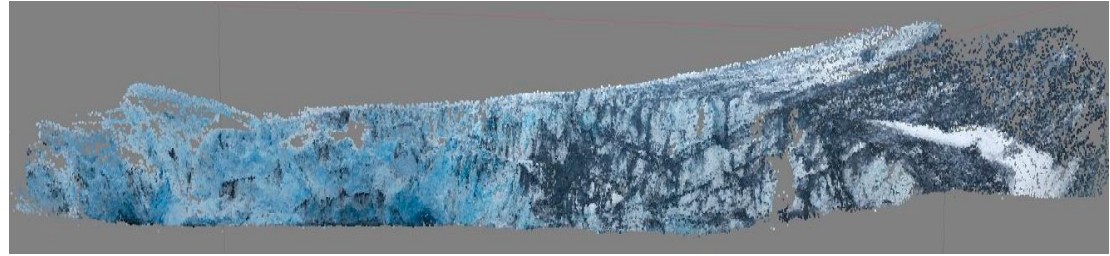

*Figure 8. Up and red, superior frontline Johnsons Glacier obtained by photogrammetric restitution. In the same figure and blue, the bottom line of intersection with the sea. Below, the resulting orthophoto the Johnsons glacier front.*

**Compiling data**

As seen above, the data obtained using photogrammetry from the Johnsons glacier front, combined with data obtained
using other methodologies (both the same compared to others) to study the temporal evolution of different fronts glaciers.

The first data are from December 26, 1957 (DOCU 1) and come from a photogrammetric flight performed metric camera to a flight altitude of 13,500 feet, with a metric camera IX Eagle Mk I and a nominal focal of 153.19 mm, which once restored using the Digi3D software, allows to obtain the position of the different glacier fronts, including the Johnsons glacier (see Table 1).

In January 1990 (DOCU 2) turned to make another photogrammetric flight, also with metric camera. In this case, using a helicopter based on the ship HMS Endurance, to a flight altitude of 10,000 feet, with a metric camera RMK A 15/23 and a nominal focal length of 153 mm. Once restored it using the same Digi3D software, the position of the different glacier fronts is obtained, including the Jonhsons glacier (see Table 2).

The photogrammetric survey of Johnsons glacier front made in 1999 (DOCU 3) with metric camera from the University
of Barcelona, is not considered since it corresponds to the upper front line and not to its intersection with the sea that is the line of interest.

In January 2010 a full image was captured by the Quickbird image system (DOCU 4) that covers the entire work area. It is a raster file format GEOTIFF UTM 20S on the ellipsoid WGS84. Its original name is 10JAN29132854-P2AS-052832138010_01_P001.TIF and was obtained from http://www.euspaceimaging.com with reference ID
"101001000B044C00". It is a black and white image with 16-bit digital values, but actually quantization levels are reduced to 11 bits. This image, was restored using ARCGIS software to get a shape file with the position of the Johnsons



glacier front (see Table 3), once the picture is rectified to a horizontal plane at sea level, using a projective transformation (Shan, 1999).

In February 2007 an image was captured by the Quickbird system (DOCU 5) which covers the entire work area (missing some insignificant Northwest rocky outcrop). It is a raster file format GEOTIFF UTM 20S on the ellipsoid WGS84. Its
original name is 07FEB03135449-M2AS-052422572010_01_P001.TIF and was obtained from http://www.euspaceimaging.com. It is an RGB color image with 16-bit digital values and an extra layer of near-infrared to 16 bits, but actually quantization levels are reduced to 11 bits. This image, was restored using ARCGIS software to get a shape file with the position of the Johnsons glacier front and Las Palmas lobe (see Table 4). This picture is rectified too to a horizontal plane at sea level, using a projective transformation.

From 2000 to 2012, the GSNCI has made scientific work on Hurd Peninsula, obtaining the position of the different glaciers fronts using GNSS techniques or theodolite. Sometimes the front was measured using a pole on the front line (Argentina, Las Palmas and Sally Rocks) with Real Time Kinematics (RTK) techniques (Blewitt, 1997). At other times, the antenna was in a backpack operator thus obtained accuracies were lower. Different shape files obtained in ARCGIS can be seen in Table 5.

| Filename | Year | $\sigma_{xy}$ (m) | Geomatic adquired method |
|---|---|---|---|
| CNDP-ESP_SIMRAD_FRONT_JOHNSON_1957.shp | 1957 | ±0,60 | Photogrammetric restitution, DOCU 1. |
| CNDP-ESP_SIMRAD_FRONT_SALLY_1957.shp | 1957 | ±1,20 | Photogrammetric restitution, DOCU 1. |
| CNDP-ESP_SIMRAD_FRONT_LAS_PALMAS_1957.shp | 1957 | ±1,20 | Photogrammetric restitution, DOCU 1. |
| CNDP-ESP_SIMRAD_FRONT_ ARGENTINA _1957.shp | 1957 | ±1,20 | Photogrammetric restitution, DOCU 1. |

*Table 1. Shape files obtained in ARCGIS for the flight made by BAS in 1957. In the third column the mean square error is shown.*

| Filename | Year | $\sigma_{xy}$ (m) | Geomatic adquired method |
|---|---|---|---|
| CNDP-ESP_SIMRAD_FRONT_JOHNSON_1990.shp | 1990 | ±2,00 | Photogrammetric restitution, DOCU 2. |
| CNDP-ESP_SIMRAD_FRONT_SALLY_1990.shp | 1990 | ±2,00 | Photogrammetric restitution, DOCU 2. |
| CNDP-ESP_SIMRAD_FRONT_LAS_PALMAS_1990.shp | 1990 | ±2,00 | Photogrammetric restitution, DOCU 2. |
| CNDP-ESP_SIMRAD_FRONT_ ARGENTINA _1990.shp | 1990 | ±2,00 | Photogrammetric restitution, DOCU 2. |

*Table 2. Shape files obtained in ARCGIS for the flight made by UKHO in 1990. In the third column the mean square error is shown.*

| Name | Year | $\sigma_{xy}$ (m) | Geomatic adquired method |
|---|---|---|---|
| CNDP-ESP_SIMRAD_FRONT_JOHNSON_2010.shp | 2010 | ±0,60 | Aereal photo, DOCU 4. |

*Table 3. Shape file obtained in ARCGIS and corresponding to the aerial photograph of QUICKBIRD system program (2010). In the third column the mean square error is shown. In this case, the image is corrected to sea level to obtain the correct planimetric position of the Johnsons glacier front.*

| Name | Year | $\sigma_{xy}$ (m) | Geomatic adquired method |
|---|---|---|---|
| CNDP-ESP_SIMRAD_FRONT_JOHNSON_2007.shp | 2007 | ±2,30 | Aereal photo, DOCU 5. |
| CNDP-ESP_SIMRAD_FRONT_LAS_PALMAS_2007.shp | 2007 | ±2,30 | Aereal photo, DOCU 5. |

*Table 4. Shape files obtained in ARCGIS and corresponding to the aerial photograph of QUICKBIRD system program (2007). In the third column the mean square error is shown. In this case, the image is corrected to sea level to obtain the correct planimetric position of the glacier fronts.*



| Name | Year | $\sigma_{xy}$ (m) | Geomatic adquired method |
|------|------|-------------------|--------------------------|
| CNDP-ESP_SIMRAD_FRONT_SALLY_2000_2001.shp | 2000 | ±0,07 | GNSS pole, DOCU 6. |
| CNDP-ESP_SIMRAD_FRONT_SALLY_2004_2005.shp | 2005 | ±0,60 | GNSS backpack, DOCU 6. |
| CNDP-ESP_SIMRAD_FRONT_SALLY_2005_2006.shp | 2006 | ±0,60 | GNSS backpack, DOCU 6. |
| CNDP-ESP_SIMRAD_FRONT_SALLY_2007_2008.shp | 2008 | ±0,07 | GNSS pole, DOCU 6. |
| CNDP-ESP_SIMRAD_FRONT_SALLY_2008_2009.shp | 2009 | ±0,60 | GNSS backpack, DOCU 6. |
| CNDP-ESP_SIMRAD_FRONT_SALLY_2009_2010.shp | 2010 | ±0,07 | GNSS pole, DOCU 6. |
| CNDP-ESP_SIMRAD_FRONT_SALLY_2010_2011.shp | 2011 | ±0,07 | GNSS pole, DOCU 6. |
| CNDP-ESP_SIMRAD_FRONT_SALLY_2011_2012.shp | 2012 | ±0,07 | GNSS pole, DOCU 6. |
| CNDP-ESP_SIMRAD_FRONT_LAS_PALMAS_2000_2001.shp | 2000 | ±0,07 | GNSS pole, DOCU 6. |
| CNDP-ESP_SIMRAD_FRONT_LAS_PALMAS_2004_2005.shp | 2005 | ±0,60 | GNSS backpack, DOCU 6. |
| CNDP-ESP_SIMRAD_FRONT_LAS_PALMAS_2005_2006.shp | 2006 | ±0,60 | GNSS backpack, DOCU 6. |
| CNDP-ESP_SIMRAD_FRONT_LAS_PALMAS_2007_2008.shp | 2008 | ±0,07 | GNSS pole, DOCU 6. |
| CNDP-ESP_SIMRAD_FRONT_LAS_PALMAS_2008_2009.shp | 2009 | ±0,60 | GNSS backpack, DOCU 6. |
| CNDP-ESP_SIMRAD_FRONT_LAS_PALMAS_2009_2010.shp | 2010 | ±0,07 | GNSS pole, DOCU 6. |
| CNDP-ESP_SIMRAD_FRONT_LAS_PALMAS_2010_2011.shp | 2011 | ±0,07 | GNSS pole, DOCU 6. |
| CNDP-ESP_SIMRAD_FRONT_LAS_PALMAS_2011_2012.shp | 2012 | ±0,07 | GNSS pole, DOCU 6. |
| CNDP-ESP_SIMRAD_FRONT_ ARGENTINA _2000_2001.shp | 2000 | ±0,07 | GNSS pole, DOCU 6. |
| CNDP-ESP_SIMRAD_FRONT_ ARGENTINA _2004_2005.shp | 2005 | ±0,60 | GNSS backpack, DOCU 6. |
| CNDP-ESP_SIMRAD_FRONT_ ARGENTINA _2005_2006.shp | 2006 | ±0,60 | GNSS backpack, DOCU 6. |
| CNDP-ESP_SIMRAD_FRONT_ ARGENTINA _2007_2008.shp | 2008 | ±0,07 | GNSS pole, DOCU 6. |
| CNDP-ESP_SIMRAD_FRONT_ ARGENTINA _2008_2009.shp | 2009 | ±0,60 | GNSS backpack, DOCU 6. |
| CNDP-ESP_SIMRAD_FRONT_ ARGENTINA _2009_2010.shp | 2010 | ±0,07 | GNSS pole, DOCU 6. |
| CNDP-ESP_SIMRAD_FRONT_ ARGENTINA _2010_2011.shp | 2011 | ±0,07 | GNSS pole, DOCU 6. |
| CNDP-ESP_SIMRAD_FRONT_ARGENTINA_2011_2012.shp | 2012 | ±0,07 | GNSS pole, DOCU 6. |

*Table 5. Shape files obtained in ARCGIS and corresponding to the GSNCI inventory data. In the third column the mean square error is shown. In this case, the shape files are for Argentina, Las Palmas and Sally Rocks lobes front. GNSS techniques are applied to obtain these data.*

| Name | Year | $\sigma_{xy}$ (m) | Geomatic adquired method |
|------|------|-------------------|--------------------------|
| CNDP-ESP_SIMRAD_FRONT_JOHNSON_2013.shp | 2013 | ±0,70 | Photogrammetric restitution, DOCU 7. |

5    *Table 6. Shape file obtained in ARCGIS and corresponding to the photogrammetric restitution from the Johnsons glacier front in February 2013. The picture was obtained with a non-metric camera.*

Finally the photogrammetric survey obtained in February 2013 with a non-metric camera (DOCU 7) for the Johnsons glacier front (see Table 6).



**Historical evolution of glacier fronts**

a)  Johnsons glacier

We operate with ARCGIS program to obtain the position of Johnsons glacier front (at its intersection with the sea) along different years, as shown in Figure 9. From an analysis of these data it follows that the front glacier advanced 74 m in the
central area (segment A) between 1957 and 1990. Then the front glacier retreated 171 m (sum of the segments A, L and F) between 1990 and 2007 to remain stable until 2010 and again to advance their central 31 m (segment L) between 2010 and 2013. Although located in 2013 ahead of his position in 2010 in this central position of the glacier front, you can see that in the south of the front changes trend and a setback of 57 m (segment J) between 2010 and 2013. Finally, note that, in the north, the glacier front has receded over 97 m between 1957 and 2013 (E segment).

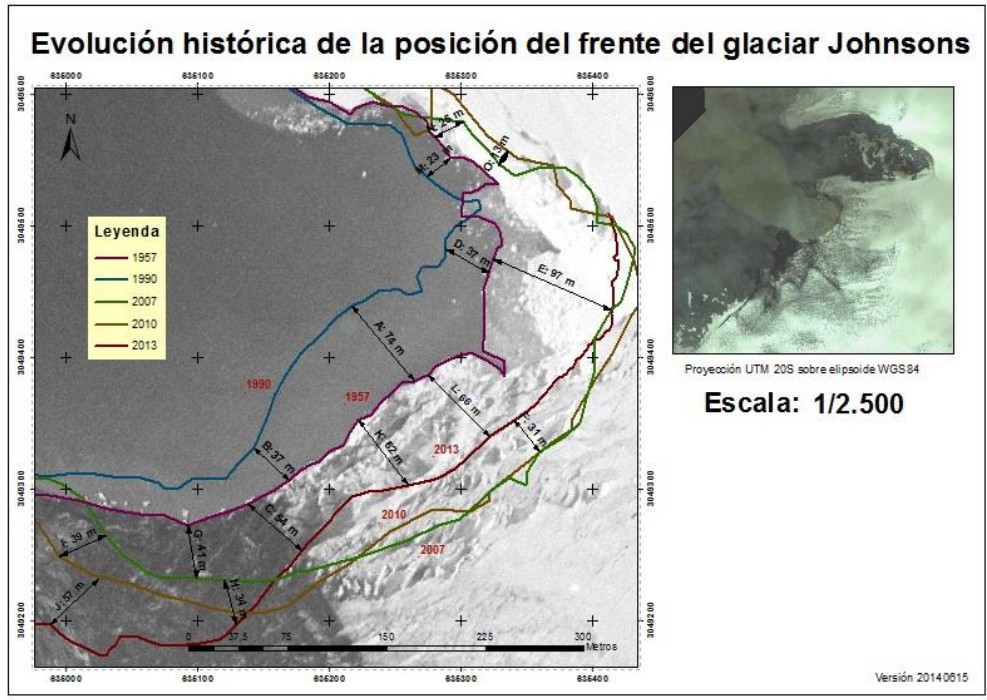

*Figure 9. History of the position of the Johnsons glacier front. The oldest front position which data are available is 1957 (in purple) and the latest for the year 2013 (in red). The base map corresponds to the aerial photograph taken in 1957 by the BAS. The scale on which the figure does not match the map scale.*

b)  Sally Rocks lobe (Hurd glacier)

Using the ARCGIS program and analyzing the position taken by the front along the years, as shown in Figure 10, it
follows that the front glacier retreated 116 m in its central area (segment A) between 1957 and 1990. Then retreated 60 m (segment B) between 1990 and 2000, to rewind back to another 47 m (C segment) in 2000-2006 and another 36 m (D segment) from 2006 to 2009. From that date, has remained stable until 2012.



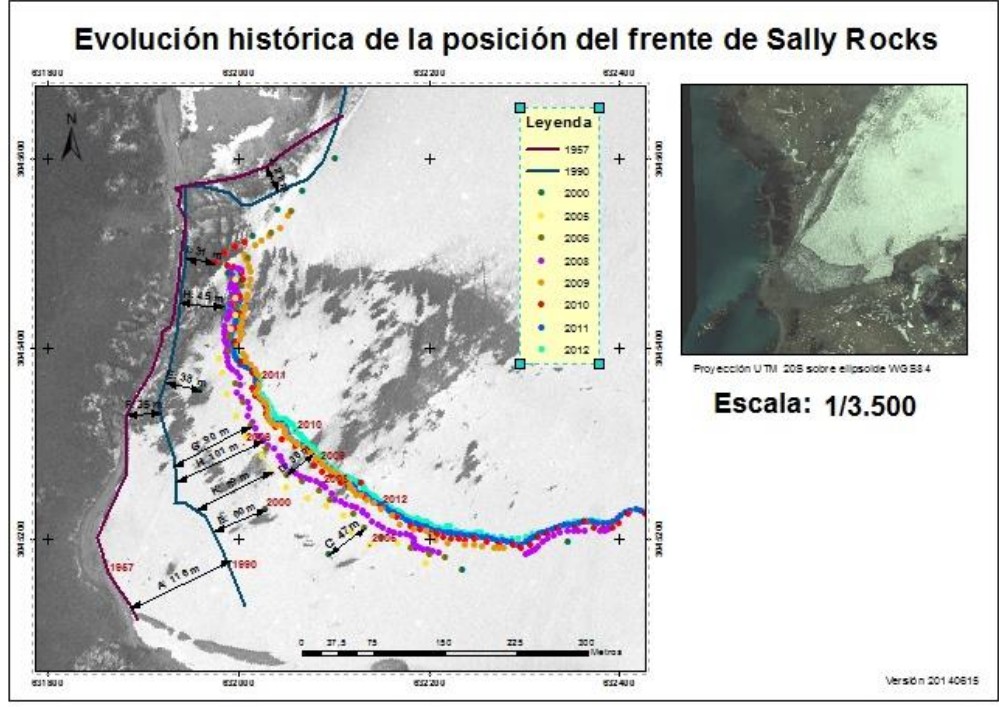

*Figure 10. Historical evolution of the position of Sally Rocks lobe front (Hurd glacier). The oldest front position which data are available is 1957 (in purple) and the latest for the year 2012 (points in cyan). The base map corresponds to the aerial photograph taken in 1957 by the BAS. The scale on which the figure does not match the map scale.*

c)  Las Palmas lobe (Hurd glacier)

As shown in Figure 11, the position taken by the front along the years follows that the front retreated 11 m in the central area (segment A) between 1957 and 1990. Then, the front retreated 24 m (segment F) between 1990 and 2000, to rewind back to another 17 m from 2000-2005 and another 14 m (D segment) from 2005 to 2007. From that date, has retreated another 10 m (although in other parts has been made) until 2009, to remain stable thereafter. Most striking in this case is that the decline experienced by the front between 1957 and 1990 is less pronounced than in the case of the front of Sally

Rocks. This is mainly because in 1957 the glacier ended at sea and had to lose weight for height, until it began to recede on the beach and accentuate its reverse speed.

d)  Argentina lobe (Hurd glacier)

Analyzed the position taken by the front along different years in ARCGIS, as shown in Figure 12, it follows that the front glacier advanced 5m in its central area (segment A) between 1957 and 1990. Then the front retreated 70 m (A + B) from

1990-2000, to rewind back to another 15 m (C segment) in 2000-2005 and another 6 m from 2005 to 2008. From that date the front has retreated another 14 m (segment I) until 2009, to remain stable after this date, with slight losses. It is striking the slight advance of the front between 1957 and 1990.





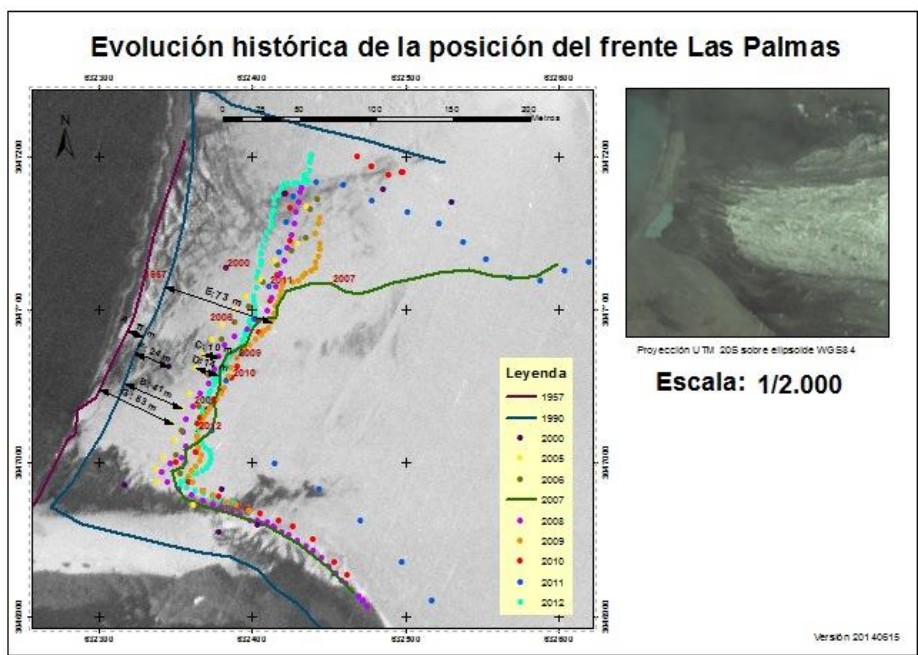

Figure 11. Historical evolution of the position of the Las Palmas lobe front (Hurd glacier). The oldest front position which data are available is 1957 (in purple) and the latest for the year 2012 (points in cyan). The base map corresponds to the aerial photograph taken in 1957 by BAS. The scale on which the figure does not match the map scale.

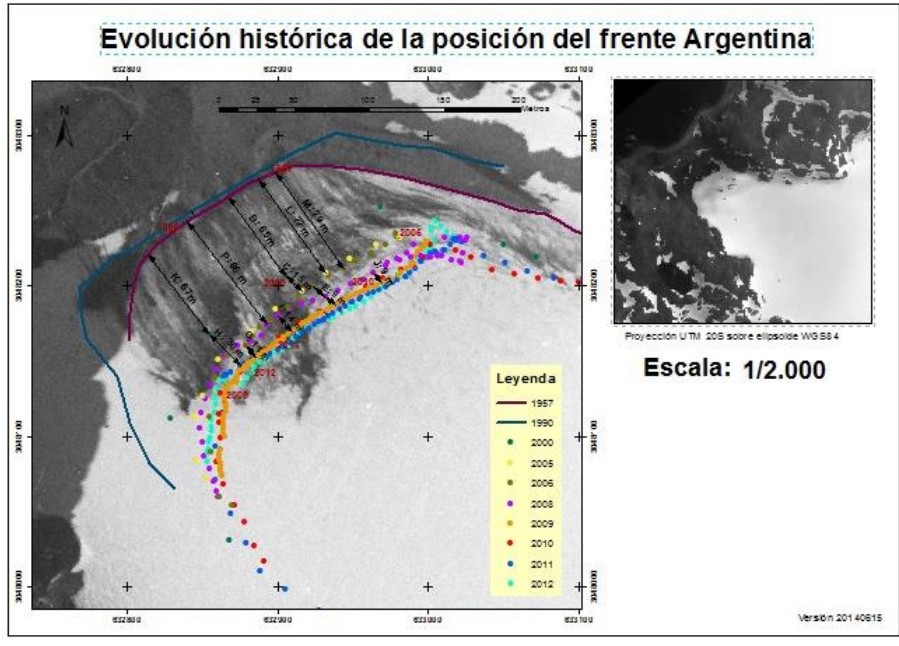

Figure 12. Historical evolution of the position of Argentina lobe front (Hurd glacier). The oldest front position which data are available is 1957 (in purple) and the latest for the year 2012 (points in cyan). The base map corresponds to the aerial photograph taken in 1957 by the BAS. The scale on which the figure does not match the map scale.



**Conclusions**

- The close range photogrammetry with non-metric cameras is a very wise for technical jobs where it is not possible to access the study area, since it implies a lowering in cost without accuracy will suffer, compared with the use of metric cameras or laser scanner systems (much higher costs). In addition, the production of different products, such as digital surface models, linear flat, bent, orthophotos, virtual three-dimensional reproductions, etc., makes the presentation of results and facilitate decision making greatly.

- Johnsons front glacier (ends at sea resulting in the production of icebergs) saw a breakthrough in the middle of his forehead between 1957 and 1990, to suffer a subsequent decrease (greater than experienced forward) between 1990 and 2010. Between 2010 and 2013, suffered a slight increase; its final position in 2013 is 140 m behind it occupied in 1990.

- Argentina lobe front (nearest from Johnsons) has a slight increase from 1957-1990, to experience a sharp decline between 1990 and 2000. From this last year, has moderated its decline until 2010, in which virtually has stabilized. The central position of the front has fallen a total of 100 m from 1990-2012.

- Las Palmas lobe front, continued setback occurs in the time period analyzed, being more pronounced between 1990 and 2009, when it seems stable. The central position of the front has fallen a total of 84 m from 1957-2012. In 1957 this glacier ended at sea and had to lose weight for height, until it began to recede on the beach and accentuate its reverse speed.

- Finally, in the case of the Sally Rocks lobe front, continued setback occurs in the time period analyzed, being more pronounced between 1990 and 2010, when it seems to stabilize. The central position of the front has fallen a total of 259 m between 1957 and 2012.

- Of all fronts studied, which lies further south (Sally Rocks) is he who has suffered a major setback. Also noteworthy indicate that the biggest drop fronts occurred between 1990 and 2010.

- Analysis of the position of the different fronts it follows that from the year 2010, the pushback suffered all has stabilized.

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
