# Peer review of "Close range photogrammetric methods applied to the study of the fronts of Johnsons and Hurd Glaciers (Livingston Island, Antarctica) from 1957 to 2013."

_Earth System Science Data, 2016_

## Referee Comment (RC1) · Anonymous Referee #1 · 26 Mar 2016

This manuscript describes a long-term data set of glacial front position changes of Johnsons and Hurd Glaciers (Livingston Island). Front positions were obtained from satellite imagery, aerial photographs, GNSS surveys and a close range photogrammetric survey in 2013.

While South Shetland Islands are one of the most easily accessible sites in Antarctica, surprisingly the glacier inventories are not updated and commonly report the ice front positions from 1957. Therefore, each credible contribution that updates those inventories and provides a detailed record of glacial extend change is highly appreciated.

[Figure]

However, in my opinion this is not the case of this study due to unsatisfying quality of both the manuscript and the presented data. Regretfully, my recommendation is to reject the manuscript.

First of all, the title "Close range photogrammetric methods applied to the study of the fronts of Johnsons and Hurd Glaciers (Livingston Island, Antarctica) from 1957 to 2013" is wrong and misleading: only a minor part of the results is obtained with close range photogrammetry. Most data comes from either aerial (1957, 1990) and satellite (2007, 2010) optical imagery or ground based GNSS surveys (2006-2012). Only a single front position obtained with close range photogrammetry is reported, as stated in the title (Johnsons Glacier 2013). Therefore, either the title or the manuscript itself needs to be changed in order to eliminate this discrepancy.

Overall, the quality of the submitted manuscript is in my opinion unacceptable for publication, basically it needs to be rewritten from a scratch. The structure of the article is inappropriate: a large portion of the text is describing the acquisition of just one front position of a single glacier (Johnsons 2013). Majority of the figures (1,3,4,7, 9-12) have embedded text in Spanish, whereas the Journal only accepts submissions in English. Large parts of the text are totally incomprehensible, it seems like it was translated from Spanish using an automatic translator. This alone would be a sufficient reason to reject the manuscript in the present form. In this case, however, I also have a major concern about the quality of the presented data:

Fig. 9 shows clearly that the ice front position of Johnsons in 1957 was erroneously delineated. Within the delimited area of Johnsons Glacier one can see icebergs and growlers floating in the sea next to the glacier termini. This is a profound error that undermines any potential trust a reader would have in the values reported in this manuscript. At this moment, there is no way to verify ice front positions in other years as unfortunately no source data is shown in figures apart from 1957 aerial photo. This could be solved by including corresponding, similar figures for other imagery used (1990, 2007, 2010) in order to ensure that such error was not repeated elsewhere.
Another concern is that a similar, more detailed work of Molina et al. (2007) is surprisingly not cited. This comprehensive study of the volume changes of Johnsons and Hurd Glaciers reports similar long-term changes for a slightly shorter period of time (1957-2000).

I have also serious doubts about the reported ice front positions of Sally Rocks lobe (Fig. 10). Authors claim that the ice front advanced "47 m (C segment) in 2000-2006 and another 36 m (D segment) from 2006 to 2009". This means front advance rates up to 12m/y, which is simply unrealistic for this glacier, given that the typical maximum ice velocity of Hurd Glacier are 4 m/y and the ice front is frozen to the ground (Molina et al., 2007). Latter work of Navarro et al. (2013) shows that over this period mass balance was generally slightly negative, I do not see a good explanation for such high advance of the ice front, apart from an error or a misinterpretation, e.g. due to a presence of snow in the forefield of the glacier.

Specific comments:

P1 l.15-20: I do not understand this part, please rewrite it

P1 l.20-23: Indicate that the non-metric camera was only used in 2013

P1 l.23-25: Major part of the work was not based on close-range photogrammetry as claimed in this sentence and in the title

P1 l.28-35: Incomprehensible, please re-write

P2 l.1-2: Satellite, not aerial imagery

P2 Fig. 1: Names of ice domes are given in Spanish, integrate them within the map, now they are outside and are hard to read.

P2 l.11-26 Please re-write this section

P3 Fig. 2: Eliminate this figure, I do not see point of explaining with a figure the concept of radial distortion

[Figure]

P3-P4 Photogrammetry is a well established method, no need to describe it in such detail. Just introduce it briefly and provide references.

P4 Fig. 3: Please translate the figure to English. Rotate the coordinate system, in this paper you consider a vertical ice wall and this figure seems to be prepared for aerial photogrammetry. Also, this and other figures are directly copied from the cited unpublished PhD thesis (Rodriguez, 2014). I guess this explains inclusion o labels in Spanish.

P5 Fig. 4: What are the black triangles? Caption states that these are ground control points at the glacier front, however they are also labelled as B1000, B2000 and B3000 bases used for theodolite measurements. I am a bit puzzled... Where are the ground control points located? Note that a numerical scale 1:6000 does not have much sense in electronic form of the article. Is this scale reported for the supposed printed size of the figure?

P5 l.18-20 What is the accuracy of theodolite measurements?

P6 l.9: Camera has 10 MP, not the lens

P6 l.10-22: This a very important section for the paper, nonetheless it is totally incomprehensible. Please re-write it.

P6 l.19-24: I like the idea of calibrating lens on such big object, of a comparable size to the studied glacier front. This allows proper lens calibration with focus set to infinity, exactly as in the subsequent field work application on Johnsons Glacier.

P6 l.36: Couldn't you simply calculate the control point position from the architectural plans of the building? Wouldn't that give you precision higher than 53 mm?

P7 l.1: How precisely was the focal length determined? Why didn't you use three different prime lenses?

P7 l.8-13: Include a table with calibration coefficients. What was their estimated error?

P8 l.5-6: This estimation is important, however the language quality makes this sentence unclear, please correct it. How was the error estimated?

P8 Fig. 6: Labels a and b in the panels and caption are missing

P8 Fig. 7: What is the name of the software?

P9 Fig. 8: Labels a and b in the panels and caption are missing. Scale bar would make this figure much more clear.

P9 l.4: How was the error estimated? Was it compared with ground control points? If so, how many? A proper assessment of the method performance is missing, for example I would expect a comparison of the calculated front position with a one obtained with a different method (high-resolution satellite imagery?)

P9 l.10: What was the rationale for repeating the photogrammetry on 1957 imagery? The ice front positions in 1957 have been already published and in fact are part of various glacial inventories. Randolph (Pfeffer et al., 2014) and Antarctic periphery (Bliss et al., 2013) inventories surprisingly were not cited. What are the differences between your results and the ones available from the inventories? Why don't you compare your results with those of Molina et al. (2007) who used the same imagery?

P9 l.20-21: I do not follow this sentence, why was this data excluded from your analysis? What do you mean by "upper front line"?

Tab.1-6: How were the mean square errors calculated? Why is the error of CNDP-ESP_SIMRAD_FRONT_JOHNSON_1957.shp two times lower than other glacier limits in 1957?

P12 l.3-9: Please correct the 1957 ice front position and change accordingly the corresponding description of ice front position changes. What was the reason for 1957-1990 advance? Why is the correct 1957 position in the region of sector E so close to the ones observed in 2007-2013? Can it be explained by the bathymetry of the embayment?

P12 Fig. 9: As stated above, the 1957 ice front position is wrong. It covers area of open water filled with icebergs (grey area near profile E). Correct front position is close to the ones for years 2007-2013. There is no way to verify ice front positions in 1990-2013 since no source data is shown. Panels should be labelled. When was the image in the smaller panel taken?

P12, l.14-17 and P13 Fig. 10: What are the black spots on the supposed glacial surface? Were they tephra layers as reported by Ximenis et al. (2000)? If so, would you associate them with any particular eruption on Deception Island? What does it say about mass balance rates of the glacier? Was the forefield snow free? If not, how was the snow surface distinguished from glacier surface? I have a major concern about the data quality: 47 meter advance in 6 years for such a slowly flowing glacier is very much, how do you explain it? The advance rate is even higher in 2006-2009 (12m/y), that sounds a bit unrealistic. Navarro et al. (2013) report negative mass balance for Hurd Glacier over this period, the snout is frozen to the bed, the maximum ice flow velocities are lower than 5m/y (Molina et al., 2007), how is it possible that the glacier front advanced 12m/y?

P13, l.14-17 and P14 Fig. 12: How do you explain the difference between 1957 and 1990 in the area to the sides of the snout? Was the forefield in 1990 snow free? Again, it is hard to verify a quality of the results without a figure with a corresponding aerial image.

P15, l.2-24: If the reported ice front positions are wrong, the discussion section has to be rewritten. Especially given that there is a serious doubt about data quality, mostly concerning position of Johnsons Glacier and Sally Rocks lobe fronts.

References:

Bliss, A., Hock, R., Cogley, J. (2013). A new inventory of mountain glaciers and ice caps for the Antarctic periphery. Ann. Glaciol. 54 (63), 191–199.

Molina, C., Navarro, F., Calvet, J., García-Sellés, D., Lapazaran, J. (2007). Hurd peninsula glaciers, Livingston Island, Antarctica, as indicators of regional warming: ice-volume changes during the period 1956–2000. Ann. Glaciol. 46 (1), 43-49.

Navarro, F., Jonsell, U., Corcuera, M., and Martín-Español, A. (2013). Decelerated mass loss of Hurd and Johnsons glaciers, Livingston Island, Antarctic Peninsula, J. Glaciol., 59, 115-128

Pfeffer, W. T., Arendt, A. A., Bliss, A., Bolch, T., Cogley, J. G., Gardner, A. S., ..., Miles, E. S. (2014). The Randolph Glacier Inventory: a globally complete inventory of glaciers. J. Glaciol., 60(221), 537-552

Rodríguez, R. (2014). Integración de modelos numéricos de glaciares y procesado de datos de georradar en un sistema de información geográfica, Tesis Doctoral. Universidad Politécnica de Madrid, pp. 235-242.

Ximenis, L., Calvet, J., Garcia, D., Casas, J.M., Sabat, F. (2000), Folding in the Johnsons Glacier, Livingston Island, Antarctica. In: Maltman, A.J., Hubbard, B., Hambrey, M.J. (eds), Deformation of Glacial Materials. Geological Society, London, Special Publications, 176, 147-157.
* * *

---

## Author Comment (AC1) · 5 May 2016

See attachment for new paper with corrections. Thanks for all.

P1 l.15-20 We re-wrote this part. P1 l.20-23: It can be indicated. P1 l.23-25: We have changed the title. P1 l.28-35: We re-wrote this part. P2 l.1-2: It has been corrected. P2 Fig. 1: The names have been changed. The base map is an Spanish map so it is impossible to change. P2 l.11-26 We re-wrote this part. Fig. 2: It has been eliminated. P3-P4 We re-wrote this part. P4 Fig. 3: The figure has been done again. P5 Fig. 4: This figure is clearest now. P5 l.18-20 This accuracy has been introduced. P6 l.9: It

has been changed. P6 l.10-22: We re-wrote this section. P6 l.19-24: Thanks. P6 l.36: We re-wrote this section and we gave an explanation for this. P7 l.1: We re-wrote this section. There are different ways to calibrate the camera and we have selected this one. The accuracy and the results are enough. P7 l.8-13: It has been included. P8 l.5-6: We re-wrote this section. P8 Fig. 6: A new figure has been included. P8 Fig. 7: It is an own software. We explain it. P9Fig. 8: A new figure has been included. P9 l.4: We have included an explanation. Landsat images and other ones have a GSD grower than 10 m so we cannot use them. P9 l.10: An explanation has been included in the paper. P9 l.20-21: We re-wrote this part. P12 l.3-9: We have included a new picture (January 1957). If it is not enough, we can change the line using this picture. P12 Fig. 9: We have included more images (1990 Sally Rocks and Johnsons) P12, l.14-17 and P13 Fig. 10: In next corrections, we can explain some of this questions. Our principal objective is to upload an inventory of data from 1957 to 2013. P15, l.2-24: After corrections, we can re-write conclusions with analysis.

Thanks for all. We have done a hard work to get all this information and it is very important to get enough quality. Thanks for your corrections.

Please also note the supplement to this comment:
http://www.earth-syst-sci-data-discuss.net/essd-2016-6/essd-2016-6-AC1-supplement.pdf

**Supplement:**

**Geomatic methods applied to the study of the fronts of Johnsons and Hurd Glaciers (Livingston Island, Antarctica) from 1957 to 2013.**

Ricardo Rodríguez[1], Julián Aguirre[2], Andrés Díez[2], Marina Álvarez[3], Pedro Rodríguez[4].

(1) Departamento de Señales, Sistemas y Radiocomunicaciones. ETSI de Telecomunicación. Universidad Politécnica de Madrid.
(2) Departamento de Ingeniería Topográfica y Cartografía. ETSI en Topografía, Geodesia y Cartografía. Universidad Politécnica de Madrid.
(3) Departamento de Lenguajes y Sistemas Informáticos e Ingeniería de Software. ETS de Ingenieros Informáticos. Universidad Politécnica de Madrid.
(4) Departamento de Matemática Aplicada. ETSI de Telecomunicación. Universidad de Málaga.

**Abstract**

The surveying of glacier fronts combines different geomatics measurement techniques. Aerial photographs and satellite images can be used for determinate the glacier terminus line. If the glacier front is easily accessible, the classic survey using total station or theodolite, GNSS (Global Navigation Satellite System) techniques, laser-scanner or close range photogrammetry are possible. When the accessibility to glaciers is not easy, close range photogrammetry proves to be useful, cheap and fast. In this paper, a methodology that combines photogrammetric methods and other techniques for the snout of the Johnsons Glacier (inaccessible) is studied. The images obtained from the front in 2013, come from a non-metric digital camera; its georeferencing to a global coordinate system is performed by measuring points GNSS support in accessible areas of the glacier front side and applying methods of direct intersection in inaccessible points of the front, taking measurements with theodolite. The result of observations obtained with different geomatics measurement techniques, were applied to study the temporal evolution (1957-2014) of the position of the Johnsons glacier front and the position of the Argentina, Las Palmas and Sally Rocks lobes (Hurd glacier).

**Link to the data repository:** http://doi.pangaea.de/10.1594/PANGAEA.845379

**Study area and previous works**

Hurd Glacier and Johnsons Glacier are located in the Hurd peninsula (Navarro et al., 2011) in the southwest of Livingston Island. Livingston Island is an Antarctic island in the South Shetland Islands, Western Antarctica lying between Greenwich Island and Snow Islands. Johnsons Glacier shows the terminus of a typical tidewater glacier, calving small icebergs into Johnsons Dock, while Hurd Glacier lobes (Argentina, Las Palmas and Sally Rocks) are land-terminating glaciers (Figure 1). Johnsons Glacier snout is continuously changing with an estimation of 50 meters per year of glacier movement near terminus area (Rodríguez, 2014). Ice velocity near terminus areas of Hurd Glacier is around 5 meters per year (Molina *et al.,* 2007).

The previous work carried with data collection fronts under study are summarized as follows:

- DOCU 1: Flight made by the British Antarctic Survey (BAS) in December 1957. A total of 5 frames (X26FID0052130, X26FID0052131, X26FID0052132, X26FID0052160 and X26FID0052161) are selected to study all of the glacier fronts.
- DOCU 2: Flight made by the United Kindom Hydrographic Office (UKHO) in January 1990. A total of 3 frames (0097, 0098 and 0099) are selected for the study of the entire glacier fronts.
- DOCU 3: Photogrammetric survey (metric camera) from the top of the glacier front Johnsons in 1999 by the University of Barcelona (Palà et al., 1999).

- DOCU 4: Satellite photograph obtained by the Quickbird system in January 2010 for the Hurd Peninsula.
- DOCU 5: Satellite photograph obtained by the Quickbird system in February 2007 for the Hurd Peninsula.
- DOCU 6: Inventory of data (2000-2012) by the Group of Numerical Simulation in Science and Engineering of the Polytechnic University of Madrid (GSNCI). These observations are made with GNSS techniques and theodolite and exclude the position of the glacier front Johnsons.
- DOCU 7: Photogrammetric survey (non-metric camera) of the front wall of Johnsons Glacier conducted in February 2013.

[Figure]

*Figure 1. Situation of the Johnsons and Hurd glaciers, location of the major landforms and situation of the Spanish Antarctic Base Juan Carlos I.*
*Base map scale 1: 25000 Geographic Service of the Army in 1991.*

**The Photogrammetry**

Photogrammetry has been defined by the American Society for Photogrammetry and Remote Sensing as the art, science, and technology of obtaining reliable information about physical objects and the environment through processes of recording, measuring, and interpreting photographic images and patterns of recorded radiant electromagnetic energy and other phenomena. The photographs are most often aerial (taken from an airborne vehicle), but terrestrial photos (taken from earth-based cameras) and satellite imagery are also used. Photogrammetry lets obtain three-dimensional information from pictures using stereoscopic vision provided by two different points of view (Wolf, 1983).

The fundamental principle used by photogrammetry is triangulation. By taking photographs from at least two different locations, so-called "lines of sight" can be developed from each camera to points on the object. These lines of sight (sometimes called rays owing to their optical nature) are mathematically intersected to produce the 3-dimensional coordinates of the points of interest. Triangulation is also the principle used by theodolites for coordinate measurement. Resection is the procedure used to determine the final position and aiming (called the orientation) of the camera when a picture is taken. Typically all the points that are seen and known in XYZ in the image are used to determine this orientation. For a strong resection, you should have at least more than ten well-distributed points in each photograph. If your measurement does not have this many points, or they are not well distributed, it is recommendable to add points (Kraus, 1993).

If the XYZ coordinates of the points on the object are known, we can compute the camera's orientation. It is important to realize that both the position and aiming direction of the camera are needed. It is not sufficient to know only the camera's position since the camera could be located in the same place but be aimed in any direction. Consequently, we must know the camera's position which is defined by three coordinates, and where it is aimed which is defined by three angles. Thus, although three values are needed to define a target point (three coordinates for its position), we need six values to define a picture (three coordinates for position, and three angles for the aiming direction). It is typical of photographs from amateur cameras that the theoretical central projection is significantly deformed by lens and film distortion. These influences can be taken into account in a bundle block adjustment by introducing correction polynomials in the observation equations, whose coefficients are determined in the adjustment. Such and adjustment is called a bundle block adjustment with additional parameters or with self-calibration (Kraus, 2007). The distortion varies with the distance of each point to the center of the optical axis. The most commonly used approach is to decompose to correct distortion in their radial components and tangential (Brown, 1971). In practice, the radial distortion $\Delta r$ is much larger than the tangential distortion so that only the first, which is expressed by the equation [1] for a point coordinate image (x, y) is ignored.

$$\Delta r = k_1 r^3 + k_2 r^5 + k_3 r^7$$
$$r = \sqrt{(x - x_0)^2 + (y - y_0)^2},$$

[1]

where $k_i$ are the coefficients of radial distortion $(x_0, y_0)$ are the coordinates of PPS in the image plane and $\Delta r$ is the radial distortion to the point considered.

Once calculated systematic errors and how to correct them and to obtain three-dimensional information from two-dimensional featuring photography, photogrammetry part of the solution that provides stereoscopic vision, in which one stage photographically recorded from two different view with a common coating, three-dimensionally can be played directly through spatial intersection producing each pairing of homologous rays appearing in both images. The procedure would be the following (Figure 2):

[Figure]

*Figure 2. Representation of point P in each of the photographs. Thus, one can calculate the unknown point coordinates in a reference system from flat photographs taken from two different points of view.*

- A terrestrial reference system $R_t$ with $O_t$ center, which is the surface you want to measure and containing the point $P$ with coordinates $(X_t, Y_t, Z_t)$ in this reference system is fixed.
- Two photographs are made from different viewpoints in this surface from $O_i\left(X_t^{O_i}, Y_t^{O_i}, Z_t^{O_i}\right)$ and from $O_j\left(X_t^{O_j}, Y_t^{O_j}, Z_t^{O_j}\right)$ (referred to $R_t$ coordinate reference system).
- Point P is represented in each of the photographs for their projections $P_i\left(X_t^i, Y_t^i, Z_t^i\right)$ and $P_j\left(X_t^j, Y_t^j, Z_t^j\right)$ (referred to $R_t$ coordinate reference system).

All this can be expressed by the so-called collinearity conditions (Kraus, 2000) which states that the center of projection, an image point and corresponding on the ground, are in the same line. The equations determining the point $P$ (whose coordinates are intended to determine), the point $P_i$ and the center of projection $O_i$ are in the same line. In like manner, the point $P$, the point $P_j$ and the center of projection $O_j$ are collinear. From the above it follows that the points $P$, $P_i$, $P_j$, $O_i$ y $O_j$ are all in the same plane, allowing us to calculate the coordinates of any point $P$ in the reference system $O_t XYZ$, by applying transformations spatial similarity (Kraus, 2007).

**Photogrammetry with non-metric camera**

To take measurements from Johnsons Glacier (February 2013), a non-metric DSLR (Digital Single-Lens Reflex) camera was used (Nikon D60). This is a typical digital camera (10 MP), not much expensive, and without excessive loss of accuracy. Obviously, its implementation requires a photogrammetric calibration, which allow to know with sufficient accuracy the internal geometry of the camera (internal orientation). Using equation [1], $k_i$ coefficients and calibrated focal length must be calculated during the calibration process.

a) Photogrammetric support

The work took place in February 2013, with fewer clouds and high visibility in the area (more than 500 m). Several control points were established making a network with permanent bases near Johnsons Glacier (see Figure 3, B1000, B2000 and B3000). In this case, these bases were measured using GNSS techniques ($\sigma_{xy} = 0,005\ m$; $\sigma_z = 0,008\ m$). In this case, we used Trimble 5700 GPS.

In addition, six control points were measured by the same technique at the end of the glacier, near lateral moraines (points P100, P200, P300, P400, P500 and P600 in Figure 3, $\sigma_{xy} = 0,011\ m$; $\sigma_z = 0,015\ m$). These points are materialized on the ground with red flags over snow. These "red points" are very easy to locate in photographs.

[Figure]

*Figure 3. Map of the B1000, B2000, B3000 bases, P100, P200, P300, P400, P500, P600 control points and I10, I20, I30, I40, I50, I60 points. In big red triangles, bases measured using GNSS techniques where theodolite was placed, in magenta triangles control points measured using GNSS techniques and finally black points (other control points) measured using direct intersection method. Line 1 and line 2 represent the track for zodiac from where the pictures were taken.*

Finally using a Wild Heerbrugg T1 theodolite, control points I10, I20, I30, I40, I50, I60 (inaccessible points) were measured. We placed theodolite above B1000, B2000 and B3000 to observe the rest of the bases, I10, I20, I30, I40, I50 and I60 to calculate control point coordinates ($\sigma_{xy} = 0,17\ m$; $\sigma_z = 0,30\ m$), using direct intersection method by resection from the three bases, allowing to have some redundancy in the observations (Domínguez, 1993). Previous to measures some pictures from the front of the glacier were taken to select optimal position for control points. These points had to be visible from bases (B1000, B2000 and B3000). T1 theodolite does not allow the measurement of distances, so only angle measurements were taken.

b)  Shooting

As already mentioned, the camera used is a DSLR camera (Nikon D60 with lens 55-200mm AF-S DX 10 MP). The only possibility of taking pictures (normal photogrammetry) perpendicular to the front of the glacier (Figure 3) is to use a zodiac (line 1 and line 2 in Figure 3) and taking images from it. Two parallel glacier front passes took place; the first at a distance of approximately 400 m (line 1), with a focal length of 95 mm and focus to infinity. The second took place at a distance of 700 m from the glacier front with the same focal length and focus to infinity. The overlap was upper than 95%, in addition to have good quality.

To grow the number of control points over the glacier, we took more pictures from a place near P500. These photographs were taken with a focal length of 130 mm and focus to infinity. In this case we mounted the camera on a tripod and we placed in the ground to take pictures using convergent photogrammetry. Then we obtained coordinates for more control points (about 20 new points). It was also observed that in these photographs appeared stakes (EJ06, EJ36 and EJ37) identified points on the glacier (measured with GNSS techniques) (Navarro *et al.* 2013).

c) Calibration

Calibration is the first phase taking place in the office, before or after making the photographic field, but respecting the same parameters. We chose for previous calibration measures a building named "Mirador", located on the street Princesa de Eboli in Madrid, which was ideal configuration for this project (Figure 4). This building also has a large open space at the front, allowing a distance similar that used in photographic shots from the Johnsons Glacier (400 meters); this distance also makes it easy to measure the corners of the windows that will be needed in the calibration grid, using a total station. For forming the corners of the reticle windows situated in a lower horizontal line, an upper horizontal line, two central horizontal lines and three vertical lines leaving a dot pattern, as can be observed in Figure 4, were chosen. We had not access to the building plans, so we decided to use this configuration to make calibration.

We chose to determine the coordinates using angular measurements because the total station did not allow to measure distances exceeding 100 m. We applied the methodology of direct intersection (Dominguez, 1993) to determinate the coordinates of all points. Observations were made from two stations simultaneously, providing one of arbitrary local coordinates for the other station coordinates and all grid points. The coordinates of the second station were obtained with an error of 12 mm. In this process we used "Leica Geoffice" software to obtain all grid points with a mean square error of 53 mm, eliminating the points whose residues exceeded this magnitude.

We calibrated the camera for three focal lengths (85, 95 and 130 mm). We used the coordinates to obtain $k_i$ coefficients, position for PPS and finally calibrated focal length using a hundred of points (see equation [1] and Table 1).

c)  Preparing images

One of the problems of work in extreme environments is that they do not allow the repetition of field observations, so a lot of errors are detected in office. It is very important to take as much information as possible during observations (of course a big number of pictures). That is why the first thing to realize is a selection of the necessary photographic peer models with an overlap of 80%.

A particular operation in the process of this project has been to correct distortion of all photographs using internal orientation parameters and generating a new set of corrected images of distortion (see Figure 5a and Figure 5b), so that may be brought into any photogrammetric program without matter what type of distortion model used.

[Figure]

*Figure 4. View of the facade of the building Mirador used for the calibration of non-metric camera used in photographic shots of Johnsons glacier front. Yellow, the measured points by using a total station.*

| Focal mm | Points | $x_0$ | $y_0$ | $k_1$ | $k_2$ | $k_3$ | $\sigma$ | $\sigma_{max}$ |
|---|---|---|---|---|---|---|---|---|
| 85,23 | 102 | 1975 px | 608 px | 1.25213889313851E-08 | -4.53330480188616E-15 | 4.48262617928856E-22 | 1 px | 15 px |
| 94,28 | 102 | 2043 px | 602 px | 1.18819659796645E-08 | -4.07129365804265E-15 | 3.78566248003642E-22 | 1 px | 13 px |
| 131,50 | 102 | 2017 px | 544 px | 1.20656410994392E-08 | -4.04403355146344E-15 | 3.68227991661486E-22 | 2 px | 20 px |

*Table 1. Coefficients of radial distortion and position for PPS. Two last columns show the mean square error for radial distortion and maximum error for radial distortion respectively. We considered 102 points for the polynomial adjustment. The first column shows the value for calibrated focal length (mm).*

10  d) Calculation of ground coordinates

With these photographs as a starting point (pictures free of radial distortion), collinearity conditions were applied, using the known support points obtained in the process of photogrammetric ground coordinates to obtain the parameters of different transformations (Helmert 3D transformations) which can obtain the ground coordinates of any point of the photographs. At the end, we obtained a total of 10 pictures to make 9 models ($\sigma_{xy} = 0,70\ m$; $\sigma_z = 0,55\ m$). Notice that

15  in this case altimetry error is less than planimetry error because XZ plane is parallel to the front of the glacier (Figure 2) so that the maximum error corresponds to planimetry.

Once all parameters are calculated from different transformations, photographs are introduced together with these parameters in an own software that allows the photogrammetric restitution, without artificial stereoscopic vision (see Figure 6) using semi-automatic correlation (Luhmann *et al.,* 2006).

20

[Figure]

| |
|---|
| *Figure 5a. Original image obtained with no metric camera.* |

*Figure 5b. The rectified image after applying the distortion function. It can be seen in the lower left corner of the image, an area affected by the radial distortion.*

[Figure]

*Figure 6. Main screen of the software developed for photogrammetric restitution. This software does not require artificial stereoscopic vision and for the restitution by locating points in both images. Clicking on an item in the frame on the left, locate the point in the right using automatic correlation. This software has been developed by these authors.*

5    The Johnsons Glacier front has two main lines for restitution. The top one of the front and the bottom line (intersection with the sea that is defined as snout). This glacier has large amount of crevasses (Figure 7b) so it is easy to use automatic correlation. With a total of 180000 points returned by automatic correlation, orthophoto is shown in Figure 7b and restitution with principal lines of crevasses is shown in Figure 7a.

[Figure]

*Figure 7a. Up and red, superior front line (Johnsons Glacier) obtained by photogrammetric restitution. Down and blue, the bottom line (intersection with the sea).*

[Figure]

*Figure 7b. Orthophoto for Johnsons Glacier front. This point cloud has 180000 point obtained by automatic correlation.*

**Compiling data**

As seen above, the data obtained using photogrammetry from the Johnsons Glacier front, combined with data obtained using other methodologies (both the same compared to others) is use to study the temporal evolution of different fronts glaciers.

The first data are from December 26, 1957 (DOCU 1) and come from a photogrammetric flight performed metric camera to a flight altitude of 13,500 feet, with a metric camera IX Eagle Mk I and a nominal focal of 153.19 mm, which once restored using the Digi3D software, allows to obtain the position of the different glacier fronts, including the Johnsons Glacier (see Table 2). Some previous work (Molina *et al.,* 2007) use these same photographs for 3D restitution but in this study, we use only the pictures to get planimetry al level sea so that the accuracy is better. Using the certificate of calibration for IX Eagle Mk I the authors have rectified photos and then, they have georeferenced photograms X26FID0052160 and X26FID0052131 using ARCGIS with an 8 parameters transformation. In fact this flight started at the end of 1956 but it was not completed (Rodríguez, 2014).

[revised manuscript text omitted]
). Near E segment the position of the front could be wrong but if we use the picture X26FID0093015 obtained at the beginning of 1957 (Rodríguez, 2014) from an incomplete British Antarctic Survey (BAS) photogrammetric flight (Figure 8a and 8b) we can see that this terminus line for the glacier is correct.

[Figure]

*Figure 8a. Terminus line for Johnsons Glacier (blue line) at the end of 1957. The base map corresponds to the BAS photogrammetric flight (date 26/12/1957).*

[Figure]

*Figure 8b. Terminus line for Johnsons Glacier (blue line) at the end of 1957. The base map corresponds to the BAS photogrammetric flight (date 19/01/1957). This flight is not completed and it only can be used for planimetry measurements at sea level.*

b) Sally Rocks lobe (Hurd Glacier)

5    Using the ARCGIS program and analyzing the position taken by the front along the years, as shown in Figure 10, it follows that the front glacier retreated 116 m in its central area (segment A) between 1957 and 1990. Then retreated 60 m (segment B) between 1990 and 2000, to rewind back to another 47 m (C segment) in 2000-2006 and another 36 m (D segment) from 2006 to 2009. From that date, has remained stable until 2012.

[Figure]

*Figure 9. History of the position of the Johnsons Glacier front. The oldest front position which data are available is 1957 (in purple) and the latest for the year 2013 (in red). The base map corresponds to the aerial photograph taken in 1957 by the BAS. The scale on which the figure does not match the map scale. The small image on the right side, corresponds to Quickbird (03/02/2007)*

    c) Las Palmas lobe (Hurd Glacier)

As shown in Figure 11, the position taken by the front along the years follows that the front retreated 11 m in the central area (segment A) between 1957 and 1990. Then, the front retreated 24 m (segment F) between 1990 and 2000, to rewind back to another 17 m from 2000-2005 and another 14 m (D segment) from 2005 to 2007. From that date, has retreated another 10 m (although in other parts has been made) until 2009, to remain stable thereafter. Most striking in this case is that the decline experienced by the front between 1957 and 1990 is less pronounced than in the case of the front of Sally Rocks. This is mainly because in 1957 the glacier ended at sea and had to lose weight for height, until it began to recede on the beach and accentuate its reverse speed.

[Figure]

*Figure 10. Historical evolution of the position of Sally Rocks lobe front (Hurd Glacier). The oldest front position which data are available is 1957 (in purple) and the latest for the year 2012 (points in cyan). The base map corresponds to the aerial photograph taken in 1957 by the BAS. The scale on which the figure does not match the map scale. The small image on the right side, corresponds to Quickbird (03/02/2007)*

d) Argentina lobe (Hurd Glacier)

5     Analyzed the position taken by the front along different years in ARCGIS, as shown in Figure 12, it follows that the front glacier advanced 5m in its central area (segment A) between 1957 and 1990. Then the front retreated 70 m (A + B) from 1990-2000, to rewind back to another 15 m (C segment) in 2000-2005 and another 6 m from 2005 to 2008. From that date the front has retreated another 14 m (segment I) until 2009, to remain stable after this date, with slight losses. It is striking the slight advance of the front between 1957 and 1990.

10    Finally Figure 13a and Figure 13b show the evolution for the snout (Johnsons Glacier and Sally Rock lobe). The base map corresponds to 1990.

[Figure]

*Figure 11. Historical evolution of the position of the Las Palmas lobe front (Hurd Glacier). The oldest front position which data are available is 1957 (in purple) and the latest for the year 2012 (points in cyan). The base map corresponds to the aerial photograph taken in 1957 by BAS. The scale on which the figure does not match the map scale. The small image on the right side, corresponds to Quickbird (03/02/2007)*

[Figure]

*Figure 12. Historical evolution of the position of Argentina lobe front (Hurd glacier). The oldest front position which data are available is 1957 (in purple) and the latest for the year 2012 (points in cyan). The base map corresponds to the aerial photograph taken in 1957 by the BAS. The scale on which the figure does not match the map scale. The small image on the right side, corresponds to Quickbird (29/01/2010)*

[Figure]

*Figure 13a. Historical evolution of the position of the snout (Johnsons Glacier). The base map corresponds to UKHO (United Kingdom Hydrographic Office, January 1990)*

[Figure]

*Figure 13b. Historical evolution of the position of the snout (Sally Rocks lobe). The base map corresponds to UKHO (United Kingdom Hydrographic Office, January 1990). For 2007 and 2010 the terminus was delimitated using GNSS techniques.*

**Conclusions**

[revised manuscript text omitted]

---

## Referee Comment (RC2) · Anonymous Referee #2 · 26 May 2016

I have read the review of anonymus referee #1 and I am afraid to say that I agree with most of the comments. I also consider necessary integrating the more developed work of Molina et al. (2007) on the subject. It is not indispensable to go through the referees' #1 comments as I think they are right. However, I strongly believe that if the authors carry out the instructions made by referee #1, the paper could be published, otherwise I would not recommend to publish it.